# Effects of Climatic Change on Soil Hydraulic Properties during the Last Interglacial Period: Two Case Studies of the Southern Chinese Loess Plateau

**Tieniu Wu [1,2]** , **Henry Lin [2]**, **Hailin Zhang [1],***, **Fei Ye [1]**, **Yongwu Wang [1]**, **Muxing Liu [1]**, **Jun Yi [1]** and **Pei Tian [1]**

1 Key Laboratory for Geographical Process Analysis & Simulation, Hubei Province, Central China Normal University, Wuhan 430079, China; wutieniu01@mail.ccnu.edu.cn (T.W.); shijunb@mails.ccnu.edu.cn (F.Y.); wangyw@mails.ccnu.edu.cn (Y.W.); liumuxing@mail.ccnu.edu.cn (M.L.); yijun@mail.ccnu.edu.cn (J.Y.); tianpei@mail.ccnu.edu.cn (P.T.)

2 Department of Ecosystem Science and Management, The Pennsylvania State University, University Park, PA 16802, USA; yongqin2004@outlook.com

* Correspondence: hailzhang@mail.ccnu.edu.cn; Tel.: +86-27-6786-7503

**Abstract:** The hydraulic properties of paleosols on the Chinese Loess Plateau (CLP) are closely related to agricultural production and are indicative of the environmental evolution during geological and pedogenic periods. In this study, two typical intact sequences of the first paleosol layer (S1) on the southern CLP were selected, and soil hydraulic parameters together with basic physical and chemical properties were investigated to reveal the response of soil hydraulic properties to the warm and wet climate conditions. The results show that: (1) the paleoclimate in the southern CLP during the last interglacial period showed a pattern of three warm and wet sub-stages and two cool and dry sub-stages; (2) when the climate was warm and wet, the soil saturated hydraulic conductivity decreased and the content of macro-aggregates increased, and when the climate was cool and dry, the soil saturated hydraulic conductivity increased and the content of macro-aggregates decreased, indicating that the paleoclimate affected both the grain size of wind-blown sediment and pedogenic process; and (3) in the soil water characteristic curves, the soil water content showed variation in peaks and valleys, indicating that the dust source and pedogenesis of the paleosol affected the water holding capacity. The findings confirmed that on the southern CLP, the warm and wet climate improved soil aggregate stability and water holding capacity, while reducing soil water conductivity. These results reveal the response of soil hydraulic properties to the climate evolution on the southern CLP, which indicate soil water retention and soil moisture supply capacities under an ongoing global warming scenario.

**Keywords:** last interglacial; Chinese Loess Plateau (CLP); soil hydraulic properties; paleoclimate

## 1. Introduction

The Chinese Loess Plateau (CLP) has historically experienced water shortages necessitating soil-water conservation. Understanding soil hydraulic properties is critical for cropland water management and soil erosion prevention. However, a wide variety of soil types is present in the CLP; besides dark loessial soil and cultivated loessial soil [1], the hydraulic properties of the widespread paleosol must also be understood.

The CLP includes dozens of paleosol and loess layers. The main sources of loess and paleosol are the northwestern inland basins [2]. The aeolian sedimentary process was accompanied by a pedogenic process. However, the pedogenic process was controlled by climatic evolution [3], and the Quaternary

paleoclimate in the CLP was characterized by large variations in cool and dry and warm and wet weather [4]. When the paleoclimate was cool and dry, the pedogenic process was relatively weak, and the visual appearance of the accumulated stratum was pale yellow, named loess; when the paleoclimate was warm and wet, the pedogenic process was relatively strong, and the visual appearance of the accumulated stratum was dark brown, named paleosol [5], so a loess-paleosol sequence recorded the Quaternary climate variation of the CLP [6]. When using the loess strata to reconstruct the paleoclimate and paleoenvironment, the thicker loess-paleosol deposits are preferable because they allow more accurate time resolution. In the plateau, the loess-paleosol strata in the northern and western part of the CLP is thicker than that in the southern and eastern parts of the CLP [5], and the thickest Quaternary loess deposit (over 400 m) was found in Lanzhou [7]. Due to the deficiency of precipitation, the pedogenesis of paleosol in northwestern CLP was weaker than in the southern CLP [8]. In the southern loess plateau, the first paleosol layer (S1) is about 8 m below the surface and developed during the last interglacial period, corresponding to Marine Isotope Stage 5 (MIS 5) [4]. This warm, wet period has been considered similar to the modern interglacial period (MIS 1) [9]. In the context of global warming, studying soil hydraulic properties under a warm, wet climate and enhanced pedogenesis is important to guide the rational use of land resources, and S1 in the southern CLP is a sensible choice.

Vertically, S1 is not an isotropic stratum. In the western, middle, and southern part of the CLP, S1 can be divided into five sub-layers: S1S1, S1L1, S1S2, S1L2, and S1S3 [8,10]. Among these sub-layers, the three strongly developed paleosol layers (S1S1, S1S2, and S1S3) developed in a warmer and wetter period, corresponding to marine isotope sub-stages 5a, 5c, and 5e. The two intervening weakly developed paleosols (S1L1 and S1L2) formed during a relatively cooler and drier period, corresponding to marine isotope sub-stages 5b and 5d, which were completely preserved [8,10,11], indicating that three strong summer monsoon events and two enhanced winter monsoon events occurred in MIS 5. These climatic events might have had certain effects on the soil hydraulic properties.

From a historical perspective, the southern CLP is a conventional agricultural field; the soil hydraulic parameters act as basic indexes for improving agricultural productivity and ecological quality. Soil hydraulic properties, such as soil moisture constant, saturated hydraulic conductivity ($K_s$), aggregate stability, and the soil water characteristic curve (SWCC), are important indexes for the growth and yield of crop plants [12]. In the Yingpan section (Yangling), some differences were found in the optimal models of the SWCC for different paleosol layers. Under high suction conditions, the water holding capacity of the fourth paleosol layer (S4) was better than that of the sub-layers of the fifth paleosol layer [13]. The soil water conditions can also be expressed by chemical elements [14]. In the southern CLP, the red ferri-argillans and migration of $CaCO_3$ and $S_r$ of S1 indicated that the paleo-climate of the last interglacial was subtropical and humid and the soil water balance was positive [15]. These results confirmed that S1 is appropriate for studying the effect of climate on pedogenesis and soil hydraulic properties.

Based on the studies described above, two issues remain pending: (1) since MIS 5 is divided into five sub-stages (MIS 5a–e), and paleosol S1 can be divided into five sub-layers, each sub-stage and sub-layer represent a specific climatic status. So, how did the paleoclimate affect the soil hydraulic parameters? (2) Did the compaction of the overlying paleosol and loess as well as the leaching process during the subsequent pedogenesis alter the soil hydraulic features in the paleosol?

To answer the above questions, the objectives of this study were (1) to determine the soil water properties of S1 profile in 10 cm intervals along a depth gradient and examine the variation trend of soil hydraulic parameters in relation to geological records and (2) to explore the connection between soil pedogenesis, soil hydraulic properties, and paleo-environmental evolution, thus addressing the climate–soil interactions since the last interglacial.

## 2. Materials and Methods

### 2.1. Site Description

The study sites were located in the southern CLP (Figure 1). This region has a warm, sub-humid continental climate, with a mean annual temperature of around 13.0 °C, and mean annual precipitation of about 650 mm, which mainly falls in July, August, and September. The vegetation above the profile is characterized by wheat, corn, secondary-growth deciduous broadleaf trees and shrubs. The top soil has seriously affected thousands of years of human cultivation, so is not suitable for reconstructing the paleo-environment and paleo-climate.

Two profiles were selected in this study: Yangling profile (YL; 34°16′60″ N, 108°02′64″ E) and Caijiapo profile (CJP; 34°20′42″ N, 107°35′38″ E). As shown in Figure 1, YL is about 80 km to Xi'an city and about 50 km to CJP town. Both of the two profiles demonstrated at least five paleosol and loess layers (Figure 2). All of the layers were well-preserved; we could detect no considerable sedimentary discontinuities or erosion relics in these two profiles. We found that the first paleosol layer (S1) is about 2.3 m thick, with a layer of Calcium carbonate nodules (0.40 m) underneath. Comparing the 2.4 m thickness of S1 in the Luochuan profile [5], ~2.5 m in the south of the CLP [16], 2.4 m in the Xunyi profile [17], and 2.1 m in the Chang'an profile [18], this stratum appears to be intact. The soil taxonomy of this paleosol is alfisol and the soil type is cinnamon soil.

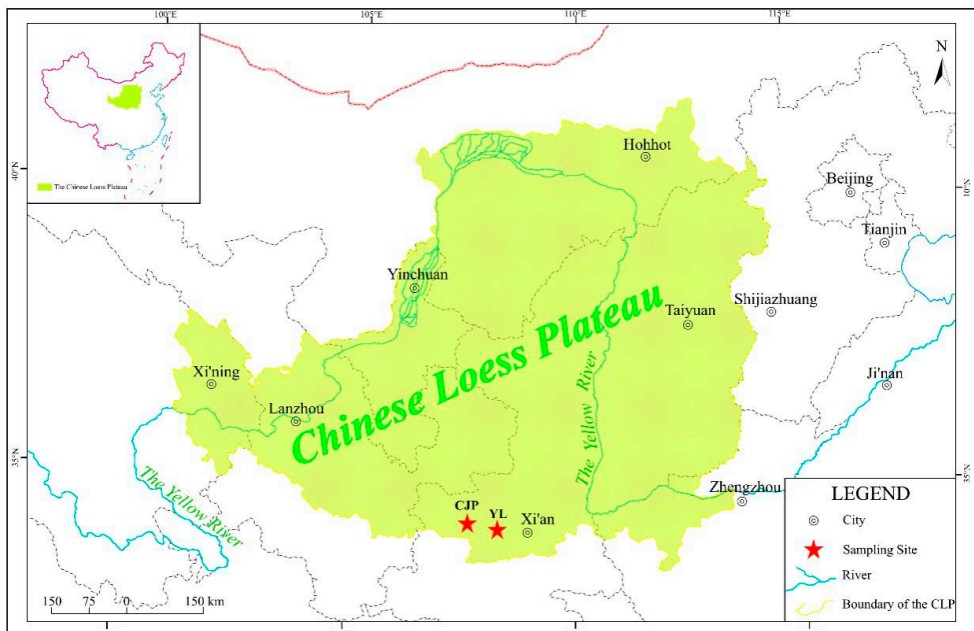

**Figure 1.** Location map of the study area. Yangling (YL) profile in the western part of Yangling District, and Caijiapo (CJP) profile north of Caijiapo town on southern Chinese Loess Plateau.

### 2.2. Soil Sampling

The two loess cliffs were cleaned by removing at least 20 cm of material, well past any signs of recent weathering or biological influence. We chose the position with intact and typical paleosol to conduct our sampling, with no visible cracks or wedges in the strata (heterogeneous filler in the column was the indication of paleo-cracks and paleo-wedges), to avoid the influence of subsequent preferential flow (occurred during the last glacial period) on soil properties. A total of 23 cutting ring (100 cm$^3$) samples were collected in 10 cm intervals from each S1 paleosol sequence, and 23 disturbed paleosol samples were obtained from the two profiles. The samples were numbered C01 to C23 (CJP section) and Y01 to Y23 (YL section) in numerical order downward from the top to the bottom of S1.

### 2.3. Laboratory Measurements and Data Analysis

Disturbed soil samples were air-dried, gently ground, and sieved according to experimental requirements for some of the subsequent analyses. Undisturbed soil samples were not air-dried, but prepared for testing the soil hydraulic properties immediately in the laboratory.

For soil particle composition, the pretreatment process involved three steps. First, 10% hydrogen peroxide ($H_2O_2$) was added to remove soil organic matter, hydrochloric acid (HCl) was added to remove carbonates, and then sodium hexametaphosphate (($NaPO_3$)$_6$) was poured into the beaker to facilitate dispersion. The grain size of the 46 samples from the two profiles was measured on a Malvern Mastersizer 3000 laser particle analyzer (Malvern Instruments Ltd., Malvern, UK) that automatically yields the percentages of the clay, silt, and sand size fractions and the median diameter of each sample, with an analytical precision of 1%. Here, we selected the median diameter and silt content (2–50 μm) as indicators of winter monsoon.

Soil organic matter (SOM) was determined as follows: About 0.5 g of air-dried and 0.25 mm sieved soil were treated with a known excess volume of chromic acid ($K_2Cr_2O_7$ + $H_2SO_4$). After the oxidation of organic carbon, the unreacted $K_2Cr_2O_7$ remaining in the contents was back titrated with standard ferrous ammonium sulphate (($NH_4$)$_2$Fe($SO_4$)$_2\cdot6H_2O$) using diphenyl Stolte amine indicator.

Hydraulic parameters could be obtained from direct laboratory and field measurements. For the SWCC, the commonly used equipment is plate extractor, high-speed centrifuge, sand box, and the HYPROP (HYdraulic PROPerty analyzer, UMS, 2015) system [19–21]. The method used to measure the SWCC depends on the texture of the soil and the magnitude of the suctions that must be established. According to previous studies, the plate extractor seemed to be more convincing than other methods for finer textured soils [22]. In this study, soil water retention at 10, 20, 40, 60, 80, 100, 300, 500, 800, and 1000 kPa was determined using plate extractors (Soil Moisture Equipment Corp., Santa Barbara, CA., USA). After the determination of the soil retention curve, all soil cores were weighed and oven-dried at 105 °C to constant weight to determine the bulk density by the core method.

For modeling the SWCCs, the van-Genuchten model was applied [23]:

$$\frac{\theta - \theta_r}{\theta_s - \theta_r} = \left[\frac{1}{1 + (\alpha h)^n}\right]^m \tag{1}$$

where $\theta$ is the volumetric water content ($cm^3\cdot cm^{-3}$), $\theta_s$ is the saturated volume moisture content ($cm^3\cdot cm^{-3}$), $\theta_r$ is the residual volumetric water content ($cm^3\cdot cm^{-3}$), $h$ is the soil water suction (kPa), $\alpha$ ($kPa^{-1}$) is the inverse of the air entry suction $u_b$ (kPa), and $m$, $n$ are the curve shape parameters (dimensionless), $m = 1 - 1/n$.

The aggregate size distributions for paleosol samples were determined by the wet sieving method [24]. Briefly, 50 g of air-dried undisturbed soil was wetted for 15 min to promote slaking of macro-aggregates. Then, the samples were placed onto a set of sieves (the mesh sizes for sieves were 5, 2, 1, 0.5, and 0.25 mm), the sieves were immersed in water and shaken vertically 30 times per minute at 4 cm amplitude for 30 min (TTF-100, Shangyu Shunlong Laboratory Instruments Factory, Zhejiang, China). The aggregates retained on each sieve were collected and weighed after drying at 105 °C for 8 h. The results are expressed as the proportion of water stable aggregates >0.25 mm diameter (also called macro-aggregates, MA). Two measurements were recorded for each sample.

The constant head method was applied to determine the saturated hydraulic conductivity [25]. Undisturbed soil samples were slowly saturated for 24 h from the bottom with deionized water. Afterward, the samples were connected to a Mariotte bottle with hydraulic head of about 5 cm to measure $K_s$ with the constant head method based on Darcy's law [26]. The outflow through the soil sample was measured in given time intervals until the flow rate stabilized. Each sample was arranged in two parallel tests.

All the physical and hydraulic parameters were analyzed in the Laboratory for Geographical Process Analysis and Simulation, Central China Normal University (Wuhan, China).

### 2.4. Chronology

For optically stimulated luminescence (OSL), samples were collected using stainless steel pipes, and OSL was conducted on four samples from 9.0 and 9.4 m depths for the CJP profile and 9.2 and 10.0 m depths for the YL profile to determine the age of the section (Figure 2). The OSL test was conducted in the Laboratory of Neotectonics and Geochronology, Institute of Geology, China Earthquake Administration (Beijing, China).

Besides OSL dating, the grain size age model [4] was also applied to determine the age of each sample according to:

$$T_m = T_1 + (T_2 - T_1)\left(\sum_{i=1}^{m} A_i^{-1}\right)\left(\sum_{i=1}^{n} A_i^{-1}\right)^{-1} \tag{2}$$

where $T_1$ and $T_2$ are the ages of the L1/S1 and S1/L2 boundaries, respectively. According to Ding et al.'s timescale [27], the ages of the two boundaries are 73 ka BP and 128 ka BP, respectively; $A_i$ is the accumulation rate at level $i$, assumed to be proportional to the >40 μm size fraction; and $n$ is the number of stratigraphic levels between $T_1$ and $T_2$.

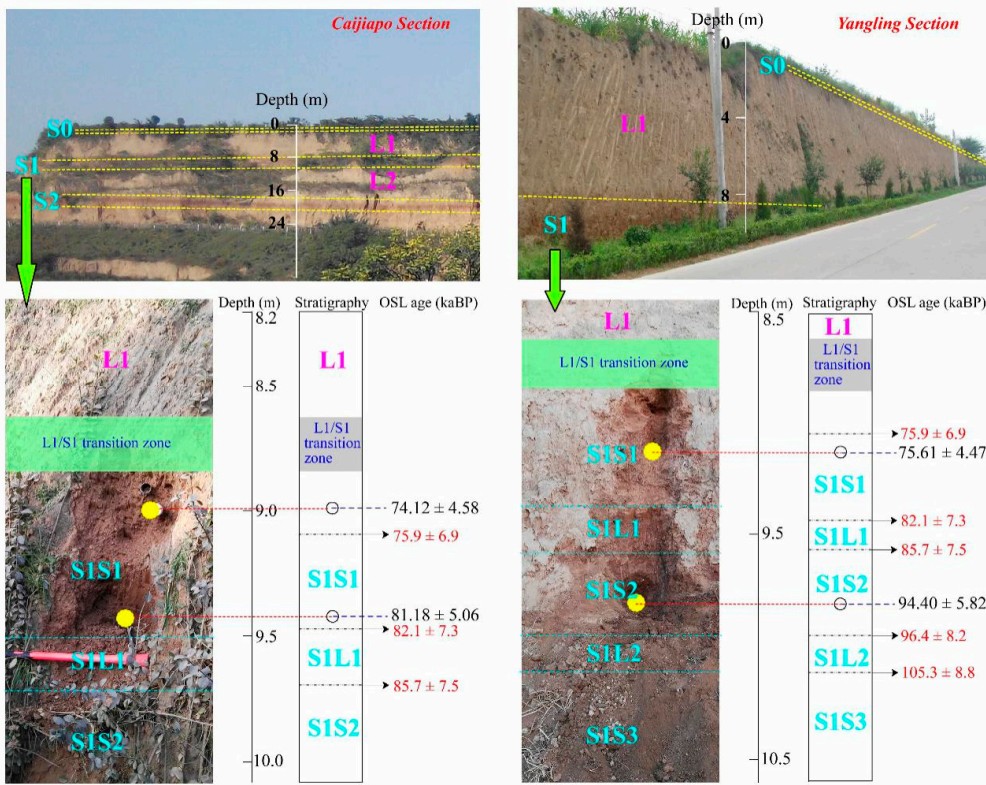

**Figure 2.** The loess-paleosol sequences at Caijiapo (**left**) and Yangling profiles (**right**). The Caijiapo section showed Holocene loess (S0), last glacial loess (L1), last interglacial paleosol (S1), penultimate glacial loess (L2), and the penultimate interglacial paleosol (S2). The Yangling section showed Holocene loess (S0), last glacial loess (L1), the last interglacial paleosol (S1). The first layer of the paleosols (S1) is divided into five sub-units: three paleosol layers (S1S1, S1S2, and S1S3) and two inter-bedded loess layers (S1L1 and S1L2) [11]. The dashed yellow lines in the upper two photos indicate the exposed paleosol layers. The optically stimulated luminescence (OSL) dating results are provided at the right of each profile photo. The OSL data, in larger font size and in black, were tested in the Laboratory of Neotectonics and Geochronology for the study profiles; the OSL data in red denote the age of S1 in the Weinan profile [28]. The thickness of sub-layers in the CJP and YL profiles are a little different from that in the Weinan profile.

As the grain size age model is widely employed in determining the loess-paleosol ages [29,30], and supported by the OSL dating results, we think that these age estimates would be reliable for the S1 level.

### 2.5. Statistical Analysis

Redundancy analysis (RDA) was adopted to analyze the effects of soil basic properties on soil hydraulic properties using CANOCO 5.0 software (Microcomputer Power, Ithaca, NY, USA). RDA is a constrained ordination method for combining multiple regression and principal component analysis. RDA can represent a multivariate data set (generally a series of samples with no less than two properties) along a decreasing number of orthogonal axes, and visualize the processed data in a two-dimensional scatter diagram, providing a relatively simple explanation of the structure of diversified data and relationships among variables [31]. Saturated hydraulic conductivity was necessary to transform the measured data using $\log(x)$. The length of projections from the arrows onto the axes indicate the contribution of the corresponding factors to the extracted axes. The cosine of the angle between the arrows reflects the correlation between the selected variables [32]. We applied the soil water RETention Curve model (RETC, version 6.02) [33] with the van Genuchten (1980) [23] model to describe soil water characteristic curves to fit the tested moisture retention data.

## 3. Results

### 3.1. Chronology of the Sampling Profiles

The OSL results are depicted in Figure 2, which confirmed that the paleosol layer we selected was correct. Compared with Kang et al. [28] who studied the Weinan profile (Figure 2, their OSL data are indicated in red), the age model we applied was credible.

### 3.2. Basic Soil Properties

By examining the basic properties of S1 (Figure 3), we found that each curve displayed roughly synchronous fluctuations. Besides the CJP and YL profiles, the other profiles in the southern CLP also recorded the analogous fluctuations during the last interglacial. Along the Weinan profile, both the magnetic susceptibility curve and the mean grain size curve showed similar variations (Figure 3V,W) [28].

Soil organic matter (SOM) content of the paleosol varies between 4.74 and 7.29 g/kg in the YL profile, and 1.74 and 8.07 g/kg in the CJP profile, with mean SOMs of the two profiles of 6.29 and 6.08 g/kg, respectively. At S1S1 and S1S2 (corresponding to MIS 5a and 5c), the SOM content is higher than that of S1Ls (Figure 3E,F), which illustrated that the vegetation cover was relatively thriving under warm and wet conditions.

Particle size distribution (PSD) is another basic feature of soil. As shown in Figure 3, in the S1Ls, the coarse particle and Md contents were higher than in S1Ss (Figure 3A–D). Compared with the benthic $\delta^{18}O$ curve, the PSD of the two profiles fluctuated synchronously with the marine records (Figure 3A–D,X). This synchronization revealed that the dust fall in the CLP was a response to the global climatic fluctuation. Particle composition is a widely accepted indicator for loess and Quaternary studies [11,34]. In addition to paleo-environmental studies, PSD was also the basis for studying the soil hydraulic properties.

### 3.3. Soil Hydraulic Properties

Although some soil hydraulic properties, such as the water retention curve and saturated and unsaturated conductivity, can be estimated by PSD [35], we measured these items using equipment to determine the difference in grain size and soil structure between each sample.

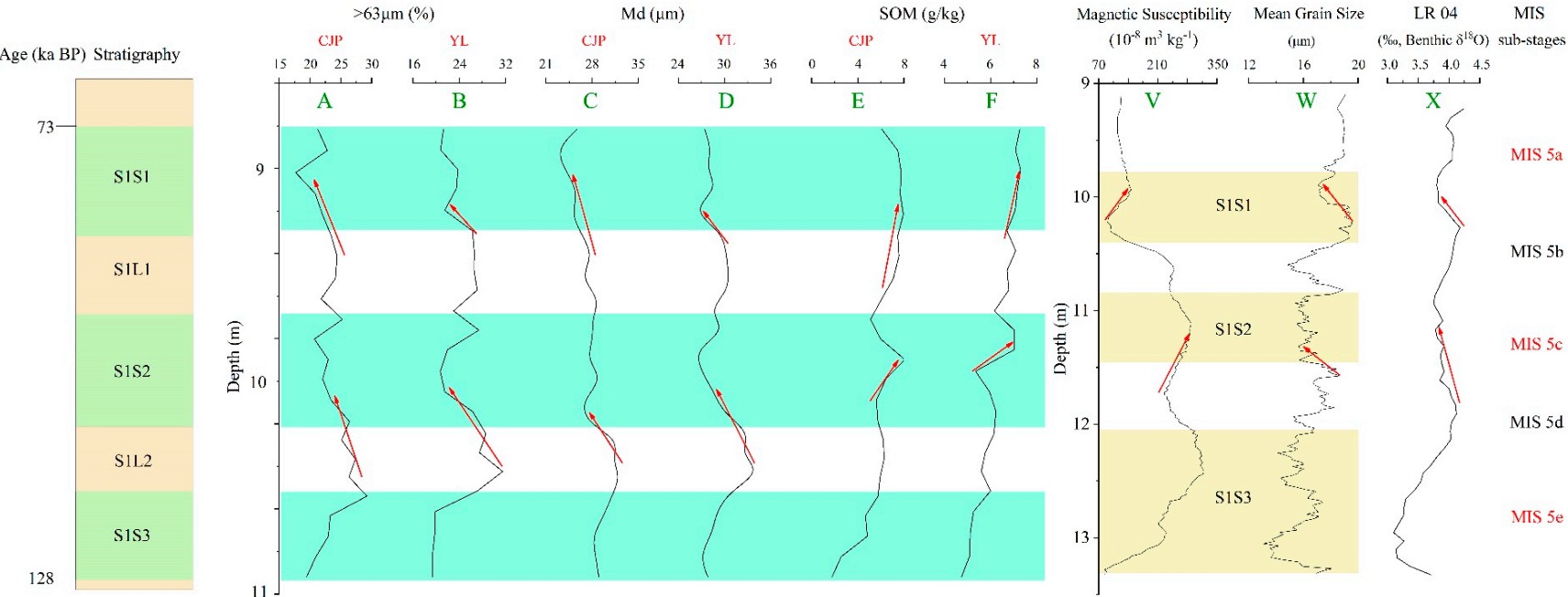

**Figure 3.** Basic properties of S1 profile as compared with the LR04 curve (X) [36] and the magnetic susceptibility and mean grain size curves of Weinan profile (V and W) [28]. In this figure, SOM is soil organic matter, MIS is Marine Isotope Stage, LR 04 is a 5.3-Ma stack of benthic $\delta^{18}$O records from 57 globally distributed sites published by Lisiecki & Raymo in 2005. The red arrows indicate the warming trend revealed by the selected parameters and marine records.

### 3.3.1. Saturated Hydraulic Conductivity ($K_s$)

The variation in $K_s$ curves G and H in Figure 4 demonstrate a fluctuation pattern similar to the LR04 $\delta^{18}$O curve. The average value of $K_s$ for the CJP profile was 1.03 cm/h, and 0.97 cm/h for YL, which could be classified as low water permeability. The $K_s$ values of S1S1, S1S2, and S1S3 were relatively lower than that of S1L1 and S1L2 (G,H, Figure 4). During the transition from cool and dry to warm and wet, the $K_s$ gradually decreased, as indicated by the orange arrows near GH in Figure 4.

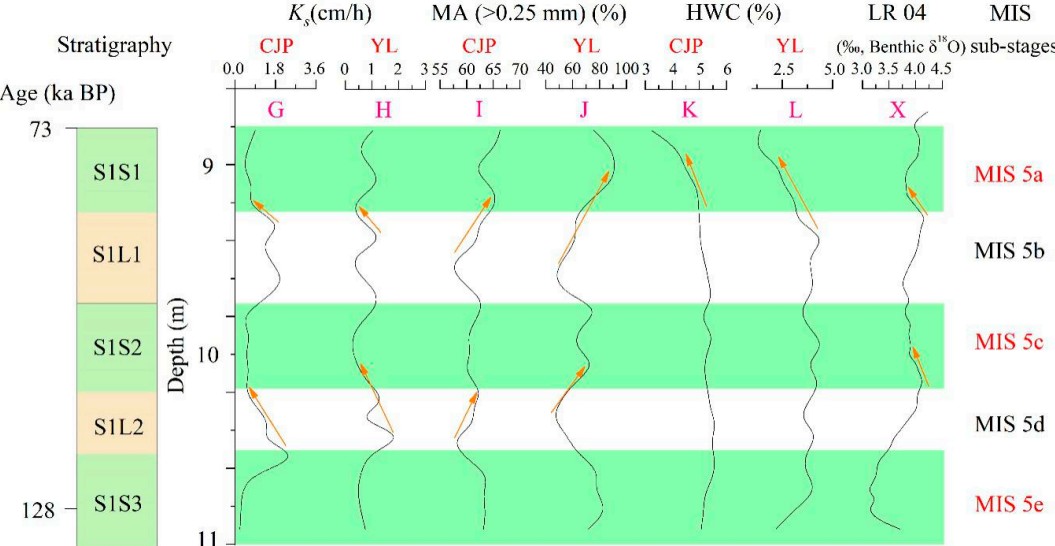

**Figure 4.** Soil hydraulic parameters of S1 profile as compared with the LR04 curve (X) [36]. The orange arrows indicate the warming trend recorded by the selected parameters and marine records. In this figure, $K_s$ is Saturated hydraulic conductivity, MA is Macro-aggregates, HWC is Hygroscopic water content.

### 3.3.2. Soil Water Stable Aggregates (Macro-Aggregates, MA)

As shown in Figure 4I,J, the curves of soil aggregates expressed a three peaks and two valleys shapes. The more the water stable aggregates, the better the soil structure, and the more suitable the soil for vegetation growth. Warm and wet was more favorable for forming soil structure than cool and dry paleo-climate.

### 3.3.3. Hygroscopic Water Content (HWC)

As shown in Figure 4K,L, the HWC of these two profiles did not fluctuate as much as the parameters above. In S1S1 and upper S1L1, the two curves showed a decreasing trend, which was contrary to the SOM variation, which might be due to increasing calcium carbonate content.

### 3.3.4. Soil Water Characteristic Curves

The SWCCs of the two profiles are shown in Figure 5. For consistency with the division of MIS 5, according to the basic physical properties, each profile was also sub-divided into five segments: samples 1–5 were assembled as a, 6–10 were assembled as b, 11–15 were assembled as c, 16–18 were assembled as d, and 19–23 were assembled as e, corresponding to S1S1, S1L1, S1S2, S1L2 and S1S3, respectively. The SWCCs of the CJP profile were marked as Ca–Ce in the left column of Figure 5, and that of YL profile were marked as Ya–Ye in the right column of Figure 5. For both profiles, the soil water holding capacity (WHC) of S1S1 (Figure 5, Ca and Ya) was the highest among all the sub-layers over the whole suction range. For CJP section, at low suction range (<100 kPa), the soil WHCs of Ca, Cc, and Ce were a little higher than those of Cb and Cd, showing a certain relationship between dust deposition, pedogenesis, and soil hydraulic properties. In contrast, at high suction range (>800 kPa), we found

no significant difference between each sample in soil WHC, so distinguishing the sub-layers by these curves was difficult. However, for the YL profile, the curves showed different patterns compared with the CJP profile. At low suction range (<100 kPa), the soil WHCs of Ya, Yc, and Ye were larger than those of Yb and Yd. At high suction range (>800 kPa), the soil WHC showed a clear periodic fluctuation pattern with higher values for Ya, Yc, and Ye and lower values for Yb and Yd, suggesting that the soil WHC of the YL profile benefited more significantly by pedogenesis than the CJP profile.

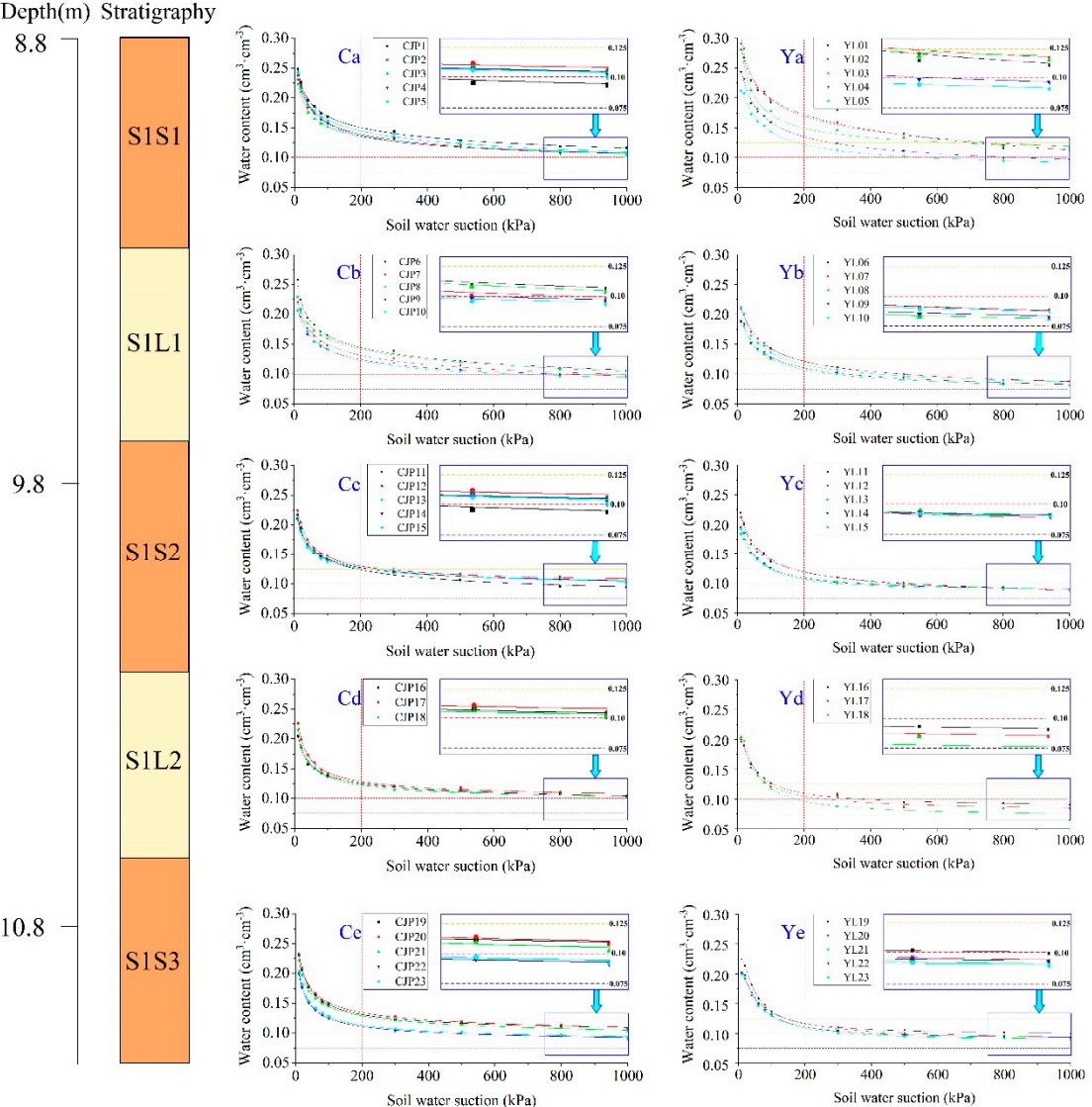

**Figure 5.** Soil water characteristic curves (SWCCs) of S1 in Caijiapo (CJP) profile (Ca–Ce) and Yangling (YL) profile (Ya–Ye). The soil water content at high suctions are magnified in blue rectangles. Taking the red dotted straight line (water content of 0.10 cm³·cm⁻³) as the reference line, the water contents under 800–1000 kPa at S1S1 and S1S3 were higher than other sub-layers of the YL profile.

*3.4. Factors Affecting Soil Hydraulic Properties*

The ordination biplots based on RDA demonstrated the correlations between soil basic properties and soil hydraulic properties (Figure 6). Here, we selected the clay content (<2 μm), total porosity, and SOM as the basic physical and chemical properties, and $K_s$, HWC, $u_b$, $\theta_s$, and $\theta_r$ as the soil hydraulic parameters. The values of $u_b$, $\theta_s$, and $\theta_r$ were calculated from the van Genuchten model [23] by RETC. For both profiles, the saturated soil water content ($\theta_s$) and saturated water conductivity ($K_s$) were positively correlated with total porosity, whereas HWC was negatively correlated with total porosity.

The residual soil water content ($\theta_r$) was positively correlated with clay content, which illustrated that soil with higher clay content could hold more water under high suction range [37]. However, the correlation between MA and clay content was not significant.

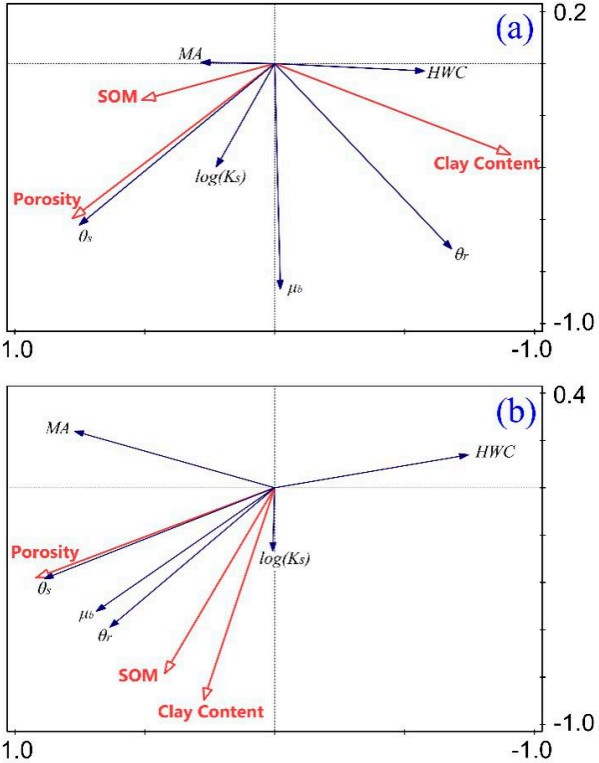

**Figure 6.** Redundancy analysis (RDA) ordination biplot showing the relationship between soil basic properties (red) and soil hydraulic properties (dark blue) in (**a**) CJP and (**b**) YL profiles. Porosity indicates total porosity; SOM, soil organic matter; HWC, hygroscopic water content; MA, macro-aggregates. The length of the arrows indicates relative importance; the direction of the arrows indicates the increase in relative content. The cosine of the angles between arrows (soil basic properties and soil hydraulic properties) indicates the degree of correlation between each pair of variables: sharp angles, positive correlations; right angles, no correlation; obtuse angles, negative correlations.

Notably, Figure 6 only shows the relationship and relative importance of the factors and indexes. For evaluating the importance of these factors, a numerical expression was needed. Table 1 describes the importance of the three factors (clay content, total porosity, and SOM) in affecting soil hydraulic properties.

**Table 1.** Results of redundancy analysis (RDA) for the soil basic properties and soil hydraulic properties.

| Profile | Factors * | Explains (%) | Contribution (%) | *p* |
|---------|-----------|--------------|------------------|-----|
| CJP | Clay content | 44.5 | 62.4 | 0.002 |
| | Porosity | 25.9 | 36.3 | 0.008 |
| | SOM | 1.2 | 0.6 | 0.57 |
| YL | Clay content | 17.6 | 25.5 | 0.002 |
| | Porosity | 49.3 | 71.5 | 0.002 |
| | SOM | 2.0 | 3.0 | 0.286 |

* SOM, soil organic matter (g/kg); Explains, to what extent each basic property explains the variation in soil hydraulic properties; Contribution, the contribution of each basic property to the soil hydraulic properties; in this study, the clay content and porosity in both CJP and YL profiles showed highly significant correlation with soil hydraulic properties ($p < 0.01$).

Among these factors, clay content and total porosity significantly impacted n soil hydraulic properties ($p < 0.01$). However, these two factors displayed some differences in terms of the importance of the two profiles. In the CJP profile, the clay content was more important than total porosity in affecting soil hydraulic properties, whereas in the YL profile, the total porosity had more influence on soil hydraulic properties than clay content. For both profiles, SOM played only a minor role in soil hydraulic properties.

## 4. Discussion

### 4.1. Climatic Evolution during the Last Interglacial

In Quaternary paleoclimate studies, the PSD of the loess-paleosol sequence in the CLP was applied to measure the strength of summer and winter monsoons [5]. In this study, the PSD of S1 in the two profiles, Weinan S1 records, and LR04 stack displayed synchronous patterns of fluctuation (Figure 3), which revealed that the paleoclimate of the last interglacial in the southern CLP experienced three warm, wet sub-stages and two cool, dry sub-stages. Besides paleoclimate reconstruction, PSD also indicates hydraulic conductivity [38]. As Figure 4 shows, the $K_s$ is relatively lower in the S1Ss than in S1Ls, which indicates that the paleoclimate not only affected the particle composition of the soil parent material, but also the soil development process, and generate various kinds of clay minerals, such as illite and kaolinite. These clay minerals could reduce the soil hydraulic conductivity [39]. In this study, the $K_s$ of the two profiles matched well with the content of illite and kaolinite in the Weinan profile [40]. S1Ss, corresponding to warm and wet periods, had more fine particles deposited due to the weaker winter monsoon, stronger soil forming process due to the enhanced summer monsoon, and more clay developed as a result of intensified pedogenesis, so the $K_s$ is lower than in S1Ls.

Local heterogeneity played an important role in soil physical properties of the two profiles. The variation of WHC in the YL profile is analogous to the PSD, $K_s$, and LR04 curves, but this phenomenon is not so obvious in the CJP profile. Besides, the MA and HWC also demonstrate some differences between these two profiles (Figure 4). For S1S1, S1S2 and S1S3, the MA in YL is larger than that of CJP, indicating the soil aggregate stability of S1Ss is stronger in YL profile; but for S1L1 and S1L2, the MA in YL is slightly lower than that of CJP, showing the soil aggregate stability of S1Ls is weaker in YL profile. In addition, the fluctuation range of HWC curve in YL is larger than that in CJP, only in the sub-layer S1S1, both curves showed a decreasing trend, in the sub-layers S1L1, S1S2, S1L2 and S1S3, the HWC curve of CJP looks almost straight, while the curve of YL oscillates notably. This differentiation might be attributed to the terrain. The YL profile is located in the river valley and the CJP profile is located on the highland. River valleys are sensitive to climate change [41]. When the paleo-climate was warm and wet, the groundwater level of the river valley was relatively high and sustained a dense plant cover, so the soil experienced a strong pedogenic process, which improved the soil water holding capacity and soil aggregate stability. When the paleo-climate was cool and dry, the groundwater level was relatively low both in the river valley and highland, weakening the pedogenic process, so the soil water holding capacity and soil aggregate stability would be weaker than what formed in warm and wet conditions. However, on the highland, the soil moisture was unrelated to the groundwater level even in the warm, wet period [42] and the vegetation was not as dense as in the river valley [43], so the soil water-holding capacity of CJP profile was not as sensitive to the climate change as in the river valley.

### 4.2. Soil Hydraulic Parameters: The Paleoenvironmental Record

As the basic and popular indexes applied in agricultural and environmental sciences [19,44], soil hydraulic parameters are rarely mentioned in Quaternary studies. Climate change and environmental evolution were important soil forming factors, providing some information about pedogenesis, and some information could be gleaned from the soil water properties. For this reason, these properties can document of climatic and environmental changes. In this study, hydraulic properties, like $K_s$ and water

stable macro-aggregates content (MA), corresponded well with the LR04 curve (Figure 4). During the relatively cool and dry period, the deposited dust was coarser and the pedogenesis was weaker, and more macro-pores (equivalent diameter > 32 μm) were produced in S1Ls [45]. The porosity directly linked to the PSD, infiltration rate and soil water holding capacity [18]. Therefore, the water holding capacity was stronger in the low suction section (<100 kPa) than in the sub-layers developed under a warm wet climate. In contrast, during the warm wet climate, the aeolian dust particles were finer (Figure 4) and the pedogenic intensity of S1Ss was stronger than that of S1Ls. Consequently, for finer-textured soils, more aggregates and micro-pores are present in the three paleosol sub-layers [46], so it had a strong water holding capacity in the high suction section. As shown in Figure 6, the RDA revealed that clay content was positively related to the residual water content ($\theta_r$). During the relatively warm and wet periods, the increased temperature and precipitation would have increased the vegetation cover, and then the dense vegetation would lead to more MA in the soil [47]. $K_s$ was also affected by the climate, but the relationship between $K_s$ and climate seems to be more complex. The strength of winter monsoons affects the particle size of dust deposition and the precipitation and temperature affect the degree of pedogenesis; both of these factors reduced the $K_s$ when the paleo-climate was warm and wet.

Soil WHC depends not only on PSD, but also on soil structure, and the soil structure was developed during the pedogenic process, some paleoenvironmental information has been preserved in the micro-structure, these information might be revealed by deeper study in WHC parameters by a combination of micro-scanning. By the way, the soil hydraulic parameters, such as MA, WHC and $K_s$, directly or indirectly relate to soil erodibility factor, which provides further support for improving soil erosion models.

*4.3. Pedogenesis and Soil Water Properties*

Pedogenesis can increase the fine-grained material over time, decreasing the $K_s$ and increasing the WHC [48]. As shown in Figure 3A–D and curves G,H in Figure 4, the $K_s$ of S1Ss was lower than that of S1Ls, and the PSD was correspondingly finer in the S1Ss. As revealed by the RDA (Figure 6), the residual water content ($\theta_r$) was significantly correlated with clay content, indicating a strong link between PSD and WHC. As mentioned above, the pedogenic intensity of the S1Ss was higher than that of S1Ls [11]. The fine particles were seemingly produced by the strong pedogenesis, which then affected the soil hydraulic properties. However, the cause–effect relationship exhibited a more complex condition in the study sections. For the paleosol S1, the parent material was not only the penultimate glacial layer (L2) deposited before the last interglacial, but also the dust accretion during the last interglacial [3]. In the CLP, the dust was transported by the winter monsoon, and the strength of winter monsoons determined the particle size of the deposits. According to previous studies, the last interglacial was a warm and wet period characterized by weaker winter monsoons, which would have led to a decrease in particle size [8]. In the context of climate warming, the finer PSD was the result of both stronger pedogenesis and weakened winter monsoons. So, during the soil forming processes, which factor was more important? More intensive work is needed to answer this question.

During the pedogenic processes, soil systems stored large amount of information about environmental factors. Such as red ferri-argillans and Sr contents, these proxies revealed that in the southern CLP, the soil water balance was positive during the last interglacial, and the southern CLP was under a humid subtropical paleoclimate [15]. This mechanism was named "soil memory" [49]. Soil water is an important soil-forming factor, it controls the material migration and biological activity. Biological activity has a significant influence on pedogenic processes, including the mixing of solid particles by soil organisms, or burrowing, bio-accumulation and tube building by animals, these processes are known as bioturbation [50]. In the CLP, the last interglacial paleosol (S1) is bioturbated with a certain amount of rounded clay balls and small burrows [8], which affects water movement and storage. Moreover, the intensity of bioturbation is controlled by the types of organisms and climate [51]. As mentioned before, the last interglacial could be divided into five sub-stages, three

warm-wet periods and two cool-dry periods, correspondingly, the bioturbation intensity should be varied with the climate fluctuations, these variations are recorded in the five sub-layers of S1 and expressed by the soil hydraulic parameters. Besides, when water passes through the soil profile, some specific traces remain [52]. As far as the paleosol is concerned, determining how extract the water flow signal from the soil properties requires further paleo-environmental study. The SWCC, $K_s$ and water stable aggregate parameters appear unsuitable for reflecting the ancient soil water flow. Soil micro-morphology and mineral composition analysis may be of assistance [53]. The next step of our study will concentrate on this issue.

### 4.4. Compaction and Soil Hydraulic Properties

We studied soil depths of 8.8–11.0 m. The overlying loess and paleosol placed enormous pressure on S1, and the compaction may have altered the soil pore geometry and the soil structure, thereby affecting the soil hydraulic properties [54]. Laboratory simulation experiments showed that the compaction affected the water retention properties for soil water potentials ranging from −5 to −80 kPa [55]; thus, the SWCCs changed significantly after compaction [56]. For the parameters in the fitting model [23], compaction resulted in the linear decrease in parameter $\alpha$ with decreasing total porosity and pore body size [57]. Although the pressure altered some physical properties, another study on the Luochuan loess-paleosol section showed that compaction was a function of synsedimentary climatic conditions [58], which implied that some relationships existed between climate evolution, compaction, and soil hydraulic properties. In this study, the synchronous fluctuation of the curves (Figures 3 and 4) proved that some soil hydraulic parameters were strongly correlated with climate variations. In other words, after long-term compaction, some soil water properties of the loess-paleosol sequence could still reflect climate evolution. Although this phenomenon seems contradictory, if we take an in-depth look, the results are in line with our perception. First, the slow and even burden from the above deposits placed all sub-layers of S1 under the similar pressure, so the difference in soil structure among each layer would be preserved as it was formed and the compressed looser layers would still be looser than the compressed tight layers, which means that some soil water parameters (such as $K_s$ and MA) could indicate paleo-climate conditions. Second, some physical properties, such as PSD, were hardly disturbed by the compaction process. Even though the buried paleosol was affected by the pedogenesis as stated above, PSD is also an important index reflecting climate variation and has been was widely used to represent the climatic rhythm [16]. The soil hydraulic parameters were affected by several factors with PSD being an important one in addition to the soil structure [59]. Based on the analysis above, we think that the soil hydraulic properties of the paleosol could also reflect the paleo-climate and paleo-environment, even if it was compacted by the above strata during the geological history.

### 4.5. Effects of Precipitation on Soil Hydraulic Properties after the Development of Paleosols

When the climate cooled and dried, the pedogenic process gradually ceased, amounts of dust accumulated on the soil surface increased, so the soil layer irreversibly transitioned into a paleosol layer. In the CLP, the paleosol S1 was covered by continuous loess deposits, which was named L1. During the loess accumulation period, the precipitation was assumed to somewhat influence the properties of S1. Heavy precipitation might produce subsurface flow. Infiltrated water moves through soils, potentially changing the soil texture, structure, and organic content, forming a preferential flow network [60]. However, at the later phase of the last interglacial (began at about 80 ka BP), the paleo-precipitation decreased gradually [61], and at the last glacial (the loess accumulation period), the paleo-precipitation was even less than during the last interglacial [62]. Therefore, the impact of the precipitation on soil hydraulic properties might be reduced. For preferential flow, studies showed that the modern water active layer was no more than 2 m on the Loess Plateau [63], whereas the modern climate of the CLP is wetter than that of the last glacial period, and the impact of water flow on soil pedogenesis is a time consuming process [52]. For the intact and typical paleosol layer, the influence of precipitation on soil

hydraulic properties may not have been so severe, and the hydraulic properties of the paleosol were mainly determined by the paleoclimate when it developed.

## 5. Conclusions

In this study, we examined some basic characteristics and soil water properties of paleosol S1 in the vertical direction in the southern CLP. We compared the LR04 benthic-$\delta^{18}$O record [35] and soil hydraulic properties. Our main conclusions are summarized as follows:

(1) The soil physio-chemical indexes revealed that the paleoclimate of the last interglacial in the study area could be divided into five sub-stages, including three warm and wet phases and two cool and dry phases. Soil hydraulic properties varied synchronously with climate fluctuation. Warm, wet climate produced more fine particles and strengthened pedogenic processes, which decreased the saturated hydraulic conductivity and increased the macro-aggregates content; cool, dry climate produced more coarse particles and weakened pedogenic processes, which increased the saturated hydraulic conductivity and decreased the macro-aggregates content.

(2) Redundancy analysis revealed that clay content and total porosity were the main factors affecting the soil hydraulic properties. In the CJP profile, the clay content played a more important role than total porosity in affecting soil hydraulic properties; in the YL profile, the total porosity more influenced soil hydraulic properties than clay content. For both profiles, soil organic matter played only a minor role in soil hydraulic properties.

(3) Although the influence could not be ignored, the precipitation since the last glacial and the compaction from the upper loess stratum (L1) did not destroy the soil hydraulic properties of S1.

Our findings confirm that the soil hydraulic properties were affected by the climatic evolution, and provide further information about land and water resources under climate change.

**Author Contributions:** Conceptualization, T.W.; methodology, T.W., Y.W. and H.Z.; software, T.W.; validation, H.L. and P.T.; formal analysis, F.Y.; investigation, T.W. and F.Y.; data curation, J.Y.; writing—original draft preparation, T.W.; writing—review and editing, H.L.; visualization, T.W.; supervision, H.L.; project administration, H.Z.; funding acquisition, M.L. All authors have read and agreed to the published version of the manuscript.

**Funding:** This research was funded by the National Natural Science Foundation of China, grant number 41701323, 41907061 and 41771261, and State Scholarship Fund of China, grant number 201806775033. This study was also partially supported by research funds of Central China Normal University from the college's basic research and Ministry of Education, grant number CCNU18QN003.

**Acknowledgments:** The authors thank Cuiting Dai and Chunlei Zhao for their help in drafting Figures 5 and 6.

**Conflicts of Interest:** The authors declare no conflict of interest.

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
