# Peer review of "Effects of Climatic Change on Soil Hydraulic Properties during the Last Interglacial Period: Two Case Studies of the Southern Chinese Loess Plateau"

_water, doi:10.3390/w12020511_

Round 1
Reviewer 1 Report
Understanding soil functioning under a changing climate is one of the key issues of earth science research. Thus this study is by all means a novel and important add-on to such attempts by reconstructing past soil evolution and hydrophysicalparameter changes from the Chinese Loess Plateau. Paleoclimatic changes with driving forcings are well known and well-documented all over the CLP for the period of the entire Quaternary; i.e. the strength of the summer and winter monsoon controls dust deposition and soil formation. Thus finding correspondence between paleo-proxies and rock physical properties of paleosols seems reasonable. This study although highlights important regional differences in hydraulic conductivity properties of the past interglacial paleosol with regards to geographical location I would appreciate to see how local heterogeneity of the soil levels influences the recorded rock physical properties at the scale of the profile perhaphs the outcrop. A brief mentioning of this in the discussion would be a great add-on to the study. The soil functioning is also somewhat oversimplified leaving out such important factors as bioturbation and the biotic component of soil evolution and functioning. Also besides PSD it would have been nice to see how the different clay minerals in the soils influence hydraulic conductivity. Nevertheless, the publication is truly outstanding and acceptable in the present form as well. A brief discussion of the importance of these parameters and how these can be involved in future studies (what proxies) would definitely improve the paper. Also if there are similar studies in China it would be worth seeing a comparison.
Author Response
Thank you very much! Your suggestions are very valuable for improving this manuscript, and we did it item by item.
This study although highlights important regional differences in hydraulic conductivity properties of the past interglacial paleosol with regards to geographical location I would appreciate to see how local heterogeneity of the soil levels influences the recorded rock physical properties at the scale of the profile perhaphs the outcrop. A brief mentioning of this in the discussion would be a great add-on to the study (Done. We added some explanations to answer this question in the second paragraph of 4.1). The soil functioning is also somewhat oversimplified leaving out such important factors as bioturbation and the biotic component of soil evolution and functioning (Done. We added bioturbation in the second paragraph of 4.3). Also besides PSD it would have been nice to see how the different clay minerals in the soils influence hydraulic conductivity (Done. We discussed the effect of clay minerals in the first paragraph of 4.1). Nevertheless, the publication is truly outstanding and acceptable in the present form as well. A brief discussion of the importance of these parameters and how these can be involved in future studies (what proxies) would definitely improve the paper (Done. We wrote a new paragraph to as the second paragraph of 4.2). Also if there are similar studies in China it would be worth seeing a comparison (Done. We added similar studies in the first paragraph of 4.2 and the second paragraph of 4.3).

Reviewer 2 Report
The paper is very interesting in the discussion of the topic, which is well connected to the concept of "soil memory". It is well organized and the contents and results are clear and supported by the analyzes.
For these reasons, I believe that the paper can be published in its current form, suggesting only to improve the quality of the images, which are not very legible.
Author Response
Response:Thank you very much! We uploaded the high-resolution figures separately. The editor will use these larger figures when publish this manuscript.
